# Formulation and Characteristics of Edible Oil Nanoemulsions Modified with Polymeric Surfactant for Encapsulating Curcumin

**DOI:** 10.3390/polym15132864

**Published:** 2023-06-28

**Authors:** Tzu-Chi Chiang, Jia-Yaw Chang, Tzung-Han Chou

**Affiliations:** 1Department of Chemical and Materials Engineering, National Yunlin University of Science and Technology, Douliu 64022, Taiwan; 2Department of Chemical Engineering, National Taiwan University of Science and Technology, Taipei 10607, Taiwan; jychang@mail.ntust.edu.tw

**Keywords:** curcumin, hydrophilic lipophilic balance number, nanoemulsions, physicochemical characteristics, encapsulation efficiency, storage stability

## Abstract

Curcumin (Cur) is a beneficial phytochemical with numerous health advantages. However, its limited solubility in oil and poor stability hinder its potential for biomedical applications. In this study, we employed a mixture of food-grade Tween 60, a polymeric surfactant, and Span 60 to adjust the hydrophilic lipophilic balance number (HLB_t_) and prepared nanoemulsions (NEs) of coconut oil (Cc oil) as carriers for Cur. The effects of HLB_t_ values, surfactant-to-oil ratio, and oil ratio on the physicochemical characteristics of the food-grade oil-NEs were investigated using dynamic light scattering, transmission electron microscopy, differential scanning calorimetry, fluorescence polarization spectroscopy, and viscometry. Increasing the addition ratio of Tween 60 in the NEs, thereby increasing the HLB_t_, resulted in a reduction in NE size and an improvement in their storage stability. The temperature and size of the phase transition region of the NEs decreased with increasing HLB_t_. NEs with higher HLB_t_ exhibited a disordering effect on the intra-NE molecular packing of Cc oil. NEs with high HLB_t_ displayed low viscosity and demonstrated nearly Newtonian fluid behavior, while those with lower HLB_t_ exhibited pseudoplastic fluid behavior. Cur was effectively encapsulated into the Cc oil-NEs, with higher encapsulation efficiency observed in NEs with higher HLB_t_ values. Furthermore, the Cur remaining activity was significantly enhanced through encapsulation within stable NEs. The biocompatibility of the Cc oil-NEs was also demonstrated in vitro. In summary, this study highlights the preparation of stable NEs of Cc oil by adjusting the HLB_t_ using Tween 60, facilitating effective encapsulation of Cur. These findings provide valuable insights for the development of Cur carriers with improved solubility, stability, and bioavailability.

## 1. Introduction

Curcumin (Cur) is a natural dietary polyphenolic compound found in turmeric rhizomes [1]. It has attracted significant interest among scientists due to its wide range of biological and pharmacological properties. These include antioxidant, anti-inflammatory, immunomodulatory, antimicrobial, anti-ischemic, anticarcinogenic, hepato-protective, nephro-protective, hypoglycemic, and antirheumatic activities [1,2,3]. Cur has also shown therapeutic effects against various diseases, such as cancer, anxiety and depression, metabolic syndrome, hypertriglyceridemia, osteoarthritis, and non-alcoholic fatty liver disease [1,4]. Despite its promising properties, curcumin faces challenges in practical applications due to its poor solubility in water and low stability in the presence of air and UV light [5,6]. It has been observed that Cur exhibits extremely low water solubility, low stability, rapid metabolism, and poor absorption, which significantly reduces its bioavailability and, consequently, its potential health benefits [7]. These limitations pose a major obstacle in effectively utilizing Cur in the fields of nutrition and biomedicine [8].

Encapsulation strategies have found extensive applications in the fields of food, cosmetics, and pharmaceuticals for effectively protecting the bioactivity of oil-soluble bioactive compounds, increasing drug solubility, and enabling controlled drug release [9,10,11]. In the pursuit of overcoming the limitations associated with Cur and enhancing its bioavailability, researchers have investigated various drug delivery systems, including liposomes [12], nanoemulsions (NEs) [13], solid lipid nanoparticles [14], vesicles [15], and polymer micelles [16]. Notably, NEs have gained significant recognition as a highly suitable approach for encapsulating Cur [17]. The incorporation of Cur into NE-based systems has demonstrated remarkable efficacy in improving its dispersion ability in nutrient products and augmenting its bioavailability [17,18]. This approach entails the addition of surfactants to the oil phase using an appropriate mechanical homogenization technique, thereby leading to the formation of a stable colloidal dispersion [19].

In order to facilitate the future commercial utilization of NE-based systems in industries such as cosmetics, health food, and pharmaceuticals, it is crucial to establish their storage stability and physicochemical characteristics [20,21]. The performance of NEs is generally influenced by their composition, with the oil component playing a critical role in the structure of the NE [22,23]. In this study, coconut (Cc) oil, known for its numerous health benefits including hypocholesterolemic, anticancer, antihepatosteatotic, antidiabetic, antioxidant, anti-inflammatory, antimicrobial, and skin moisturizing properties [24,25], was selected as the core oil for the preparation of NEs. Additionally, the surfactant is another major component in the Cc oil-NEs due to its ability of reducing interfacial energy for colloidal stabilization. The use of a single surfactant may not be sufficient to achieve stable emulsions, and the combination of two or more surfactants often exhibits superior performance in various dispersion systems [26,27,28]. Sorbitan monostearate (Span 60) and polyethylene glycol sorbitan monostearate (Tween 60) are moderate surfactants widely employed in the cosmetic, food, and pharmaceutical industries [29,30]. Tween 60 and Span 60 are food-grade non-ionic surfactants that have been approved by the Food and Drug Administration (FDA) [31]. They are known for their stability, which is crucial for ensuring the long-term shelf life of formulated products. Span 60 and Tween 60 can withstand a wide range of temperatures and pH conditions without undergoing significant degradation or losing their emulsifying properties. Therefore, in this study, Cc oil-NEs were designed using a combination of Span 60 and Tween 60 as surfactants to obtain stable nanodispersions.

Non-ionic surfactants consist primarily of a hydrophilic head group and a lipophilic tail, and their balance is quantified by the hydrophilic–lipophilic balance (HLB) number. The HLB number of non-ionic surfactants can be theoretically calculated [32]. In the case of binary mixed surfactants, the total HLB number (HLB_t_) can be determined at a constant temperature using the following equation [32,33]: *HLB*_*t*_ = *w*_1_ × *HLB*_1_ +*w*_2_ × *HLB*_2_(1)
where *w*_1_ and *w*_2_ represent the weight fractions of the individual surfactants. *HLB*_1_ and *HLB*_2_ correspond to the HLB numbers of each surfactant. Span 60 has a relatively low hydrophilic–lipophilic balance number (HLB_t_) value, approximately 4.7, while Tween 60 has a higher HLB value of around 14.9 [34]. The difference in HLB values between Span 60 and Tween 60 allows for a broader range of applications and formulation possibilities. Therefore, in this investigation, we utilized a mixed surfactant system of Span 60 and Tween 60 with HLB_t_ values ranging from 4.7 to 14.9. 

In this study, Cc oil-NEs were prepared using a combination of homogenization and ultrasonication methods to encapsulate Cur. The primary objective was to examine the physicochemical, rheological, and encapsulation properties of Cc oil-NEs with varying compositions. Various analytical techniques, such as dynamic light scattering, transmission electron microscopy (TEM), differential scanning calorimetry (DSC), fluorescence polarization spectroscopy, viscometry, and high-performance liquid chromatography (HPLC), were employed to comprehensively characterize the Cc oil-NEs. By varying the ratios of Span 60 and Tween 60, the *HLB_t_* values were adjusted to investigate the impact of the *HLB_t_* on the performance of Cc oil-NEs. Furthermore, the in vitro biocompatibility of the NEs was assessed using the 3-[4,5-dimethylthiazol-2-yl]-2,5-diphenyl tetrazolium bromide (MTT) assay. 

## 2. Materials and Methods

### 2.1. Materials

The water phase used was pure water with a resistivity of 18.2 Ωcm from a Milli-Q-plus purification instrument (Burlington, MA, USA) for the preparation of all NEs. The Cc oil (Mariox Resourse, Kuala Lumpur, Malaysia) was purchased from a local Taiwanese company (Yunlin, Taiwan). Span 60, Tween 60, and 1,6-diphenyl-1,3,5-hexatriene (DPH) with a purity of ~98% were obtained from Sigma-Aldrich (Steinheim, Germany). Cur with a purity >97.0% was obtained from Tokyo Chemical Company Co., Ltd. (Tokyo, Japan). Methanol was purchased from Avantor Performance Materials Inc. (Center Valley, PA, USA). Ethanol was obtained from Echo Chemical Co., Ltd. (Miaoli, Taiwan). Uranyl acetate was obtained from Electron Microscopy Science (Hatfield, Britain).

### 2.2. Cc Oil-NE Manufacture

The manufacture procedure of Cc oil-NEs was as described in previous works with a slight modification [28,35]. Cc oil, mixed Span 60/Tween 60, and water phase were pre-heated to 70 °C in 5 min. Then, the oil phase including surfactants was added to the water phase with homogenization at 12,000 rpm for 4 min using a D-500 homogenizer (Wiggen Hausser, Germany). Furthermore, the emulsified sample was sonicated under a power of 70 W at 45 °C for 30 min using an ultrasonicator (Q700, Qsonica, LLC., Sarasota, FL, USA). Various NE formulations were prepared with oil weight ratio of 1~8%, surfactant to oil ratio (SOR) of 0.5~2, and HLB_t_ of 4.7~14.9. Each NE formulation was repeated at least three times (*n* = 3).

### 2.3. Size, Polydispersity Index (PdI), Zeta Potential (ZP), and Physical Stability of NEs

Average particle size (APS), PdI, and ZP of Cc oil-NEs were measured by Zeta Plus Analyzer (Brookhaven Instruments Corporation, Holtsville, NY, USA) based on the principles of dynamic light scattering for size and distribution and Doppler velocimetry (electrophoretic light scattering) for zeta potential. Before each measurement, the fresh samples were suitably diluted in pure water to avoid light interruption. The values of APS, PdI, and ZP (the mean ± standard deviation) were calculated from triplicate measurements per sample. Additionally, the NEs, with a volume of 20 mL, were stored at room temperature, approximately 25 °C. The storage stability the of NEs was evaluated based on the following criteria: APS < 500 nm, PdI < 0.4, or no optical phase separation observed. 

### 2.4. TEM Observation of NEs

A transmission electron microscope (Hitachi H-7500, Tokyo, Japan) was utilized to visualize images of Cc oil-NEs. For sample preparation, 5 μL of NEs was spread onto a 3 mm copper grid coated with carbon and left to stand for 10 min. Excess sample was removed using filter paper and 2 μL of 2.5% (*w*/*w*) uranyl acetate solution was dropped onto the sample grid for 2 min. The stained NE was subsequently kept in a drying oven for 12 h. Finally, a TEM operation was performed at 100 kV to capture the sample image.

### 2.5. DSC Measurement of NEs

Thermal transition characteristics of Cc oil-NEs without lyophilization were directly measured by DSC (DSC 1 instrument, Mettler Toledo Inc., Schwerzenbach, Switzerland) combined with an immersion cooler (TC45, Huber USA Inc., Raleigh, NC, USA). First, 20 µL of the sample was added into an aluminum sealed pan and 20 µL of pure water in the reference pan. The two pans were put into the DSC chamber with an initial temperature of 25 °C. The thermal analysis of NEs was performed using three heating–cooling cycles with a 5 °C/min rate under a temperature range of −10 °C to 80 °C.

### 2.6. Fluorescence Polarization of NEs 

The fluorescence polarization technique has been widely utilized to investigate molecular mobility and interactions in lipid vesicular systems [36]. In recent years, this approach has been applied to NE systems for realization of their intra-particle behavior [37,38]. To begin with, a hydrophobic marker, DPH, was added into the oil phase with a 1:2000 weight ratio of the marker to oil. After that, samples were prepared by the homogenization–sonication previous mentioned. The fluorescence polarization value (P) of NEs was determined on a fluorescence spectrometer with its accessory soft system (Perkin Elmer LS 55, Shelton, CT, USA). The measurement and calculation of P were carried out as reported before [36]. 

### 2.7. Rheological Properties of NEs

Rheological properties of NEs were measured by a Brookfield DV-Ⅲ Ultra Programmable Rheometer, (Brookfield engineering laboratories Inc., Middleboro, MA, USA). The relationship of shear stress varying with shear rate for NEs was measured using a cone–plate model, and the cone (CPA-40Z) had a diameter of 4.8 cm with a tilt angle of 0.8°. The NEs with various compositions were loaded into the instrumental chamber and pre-equilibrated at 25 °C for 20 min before each measurement.

### 2.8. Cur Encapsulation of NEs

Amounts of Cur encapsulated by NEs were assayed by a high-performance liquid chromatography (HPLC) instrument. After centrifugal ultrafiltration, the encapsulation efficiency (EE) of Cc oil-NEs was obtained by the following equation:EE (%) = (total Cur amount − non-trapped Cur amount)/total Cur amount × 100%(2)

The HPLC system was equipped with an L-2200 auto-sampler, L-2130 pump, L-2455 diode array detector (Hitachi, Ltd., Japan), and a temperature controller (SUPER CO-150, Enshrine Company, Kaohsiung, Taiwan). The RP18 column, Microsorb-MV 100-5 (100Å, 5 µm, 4.6 × 250 mm, Agilent, Santa Clara, CA, USA), was controlled at 35 °C. Pure methanol was used as the mobile phase and its flow rate was 1.0 mL/min. The detection wavenumber ranged from 200 nm to 700 nm and was used with a Cur-specific wavenumber of 425 nm. NEs encapsulating Cur were placed on a 5K MWCO membrane (Vivaspin 500, Sartoius Stedim Lab Ltd., Gloucestershire, UK) and then centrifuged for 2 h at 8000 rpm, 4 °C (Centrifuge 5424R, Eppendorf, Hamburg, Germany). After that, the amount of non-trapped free Cur from NEs was determined by HPLC as described above.

### 2.9. Cur Stability Analysis 

The chemical stabilities of Cur dispersed in water and Cur entrapped in NEs were estimated by using a UV–Vis spectrophotometer (SPECTROstar^®^ Nano BMG LABTECH GmdH, Ortenberg, Germany). The activity change in Cur was measured by comparing the sample absorbance at 425 nm with the initial one in the designed period.

### 2.10. Cytotoxicity Assay for NEs

To assess the in vitro biocompatibility of Cc oil-NEs emulsified using a mixture of Span 60 and Tween 60 with different HLB_t_ values, the toxicity of these formulations on human premalignant keratinocytic (HaCaT) cells was evaluated using a 3-[4,5-dimethylthiazol-2-yl]-2,5-diphenyl tetrazolium bromide (MTT) assay as modified from a previous report [36]. NEs with different HLB_t_ values at a fixed SOR of 1 and an oil concentration of 2% were used as stock solutions and diluted to different concentrations (0.1, 0.01, 0.001, 0.0001 mg/mL) by adding an appropriate amount of culture medium. In the control condition, the control groups were treated with pure water in the same volume as the NEs. After treating the cells with samples for 24 h, the culture medium was replaced with fresh medium, and the cell survival was measured at an absorbance of 570 nm using a multi-functional microplate reader (SPECTROstar^®^ Nano BMG LABTECH GmdH, Ortenberg, Germany). Cell viability was defined as the percentage of average surviving cells relative to the control groups. 

### 2.11. Statistical Analysis

All the experiments were repeated at least three times in this work. The data were expressed as the average value ± standard deviation. The results herein were statistically analyzed with one-way ANOVA by using Design-Expert 10 software (Stat-Ease^®^, Minneapolis, MN, USA), where *p* < 0.05 was considered as significant.

## 3. Results and Discussion

### 3.1. Size, Zeta Potential, and Storage Stability of Cc oil-NEs

The average particle size (APS), polydispersity (PdI), zeta potential (ZP), and stability duration of Cc oil-NEs emulsified with Span 60 are presented in Table 1. However, when Cc oil was emulsified solely with Span 60, the resulting NEs were not stable for extended periods, with stability durations of ≤12 days. Additionally, when the oil ratio was ≥4% and SOR was ≥1.5, stable NEs could not be formed using Span 60 and Cc oil in water, with stability durations of less than 1 day. On the other hand, Table 2 displays the APS, PdI, ZP, and stability duration of Cc oil-NEs emulsified solely by Tween 60. These formulations, regardless of the oil ratio and SOR, resulted in the formation of stable NEs. Furthermore, these NEs exhibited enhanced stability and smaller particle size compared to those prepared with Span 60. From a molecular structural perspective, Tween 60 possesses a hydrophilic polyethylene glycol (PEG) side chain, which contributes to a larger polar headgroup when compared to Span 60. This structural attribute exerts a steric effect on particles, thereby impeding their aggregation and inducing a high curvature of colloids, leading to the formation of smaller NEs. Similar effects, including reduced particle size and enhanced stability, have been documented upon the incorporation of other PEGylated surfactants into nanodispersions [36,39,40].

Table 3 lists the APS, PdI, ZP, and stability duration of Cc oil-NEs with varying HLB_t_ and SOR while maintaining a constant oil ratio. When HLB_t_ ≧ 9.8 and SOR < 2, the size of Cc oil-NEs decreased, and their storage stability increased as the SOR increased. Among the range of HLB_t_ from 4.7 to 14.9, NEs with SOR = 2 showed the largest size and the lowest stability. On the other hand, at a fixed SOR, the storage stability of Cc oil-NEs improved with increasing HLB_t_. This indicates that a higher proportion of Tween 60 in the surfactant mixture can extend the stability duration of Cc oil-NEs at a constant SOR and oil ratio. The ZP of Cc oil-NEs was negative and did not show significant changes with SOR when HLB_t_ was kept constant. However, NEs with an HLB_t_ of 14.9 exhibited a smaller absolute value of ZP compared to NEs with other HLB_t_ values. The negative surface charges observed in Cc oil-NEs can be attributed to the presence of fatty acids in Cc oil. Although the large PEG group in Tween 60 might influence the surface charge of the NEs, it also contributes to the improved stability of Cc oil-NEs.

### 3.2. Morphology of Cc Oil-NEs 

TEM is a widely accepted technique used to directly visualize the morphology and size of NEs [41,42]. Figure 1 displays TEM images of Cc oil-NEs with varying HLB_t_ values while keeping oil ratio and SOR constant. The images reveal that the designed formulations of Cc oil-NEs exhibited a predominantly circular shape. The particle size observed in the TEM images aligns with the size distribution range obtained from DLS measurements. Some irregular nanoparticles and aggregations were observed, which could be attributed to water evaporation during the drying process. Moreover, an increase in HLB_t_ leads to a reduction in particle size, as evident from the TEM images. This trend can be attributed to the higher proportion of Tween 60 with a PEG group, resulting in an increased surface density of PEG on the NEs. Similar observations have been reported in other studies concerning nanoparticles [36,43].

### 3.3. Effects of Surfactant Amount and HLB_t_ on Phase Behavior of Cc Oil-NEs

The thermal behavior of NEs, which is closely related to their physical performance and drug release characteristics, can be evaluated using the DSC technique [35,44]. Figure 2 presents that DSC thermograms of Cc oil-NEs emulsified with Span 60 at different SOR values and oil weight percentages of 1% and 2%. Each heating curve exhibited two distinct phase transition peaks: the α phase which occurred between 10 °C and 30 °C and the β phase which occurred between 42 °C and 60 °C. The occurrence of two phases transition peaks indicates the existence of two different molecular packing arrangements within these NEs. When the oil ratio is held constant, the β peak shifted to higher temperatures and showed a slight increase in peak size with an increase in SOR. This suggests that the inclusion of higher amounts of Span 60 may enhance the attractive forces between molecules within the NEs, especially in the rigid phase. Conversely, increasing the oil content of the NEs resulted in a slight increase in peak size without a noticeable shift in the peaks when the SOR value was kept constant. This observation implies that a larger quantity of oil in the NEs requires a higher amount of endothermal heat during the phase transition.

Figure 3 illustrates the DSC thermograms of Cc oil-NEs emulsified with pure Tween 60 at various SOR values and oil ratios of 2% and 4%. In comparison to Span 60 at the same oil ratio and SOR, only the α phase transition region was observed and peak sizes were smaller. This indicates that Tween 60 had a disruptive effect on the molecular packing of Cc oil-NEs, which can be attributed to the presence of the PEG group in its structure. Similarly, the disorder effect of PEG has also been observed in the lipid vesicular systems [36]. Additionally, when twice the amount of oil was added to the NEs, an increase in the peak size of the α phase was observed, resulting in an increase in the heat capacity during the phase transition. Furthermore, Figure 4 demonstrates the influence of HLB_t_ on the DSC curves of Cc oil-NEs with a fixed SOR and oil content. As the HLB_t_ increased, both the α and β peak sizes decreased, and their phase transition temperatures shifted to lower values. This indicates that an increase in the ratio of Tween 60 resulted in a loosening effect on the molecular arrangement of the NEs. This effect could be attributed to an increase in the surface density of the PEG group, as both Span 60 and Tween 60 have the same monostearate structure.

### 3.4. Effect of HLB_t_ on the Intra-NE Molecular Mobility 

Recently, fluorescence polarization analysis has emerged as an approved technique for assessing molecular mobility within nanoscale colloids [35,40,45,46]. Nevertheless, there have been no detailed studies on the molecular mobility of Cc oil-NEs influenced by HLB_t_ to the best of our knowledge. Figure 5 illustrates the variation in DPH polarization values of Cc oil-NEs with different HLB_t_ at a fixed oil ratio and SOR. The polarization value of NEs emulsified with Span 60 was nearly twice that of NEs with Tween 60, indicating a rigidified effect on the intra-NE molecular mobility provide by Span 60. Additionally, a depolarization characteristic was observed, reaching a plateau with increasing HLB_t_. This suggests that increasing the Tween 60 ratio in this mixed binary surfactant could enhance molecular mobility in the Cc oil-NEs. However, it is important to note that this effect may become limited when the HLB_t_ value exceeds 9.8. The increase in molecular mobility may be associated with an increase in the number of PEG groups on the surface of the NEs, which is supported by the aforementioned DSC results. The presence of a higher quantity of PEG groups may introduce greater flexibility and mobility within the NEs, leading to the observed depolarization effect. This phenomenon is consistent with findings in liposomal systems, where the molecular mobility of liposomal membranes was found to be influenced by the amount of incorporated PEG-lipid [47].

### 3.5. Influences of Surfactant Amount and HLB_t_ on Rheological Properties of Cc Oil-NEs

The rheological properties of NEs play a crucial role in their physical stability and their suitability for specific applications [48,49,50,51]. Shear stress–strain rate flow curves are fundamental and widely used measurements for assessing various rheological characteristics [52,53]. Figure 6 depicts the influence of SOR and HLB_t_ on shear stress versus shear rate profiles of Cc oil-NEs at a constant oil percentage. It can be observed that the slope of the curve, which represents viscosity, increased with an increase in SOR at a constant HLB_t_, particularly at low HLB_t_ (7.76). The additional amount of surfactant contributes to a stronger van der Waals force between molecules in the NEs, leading to higher viscosities. This trend was more pronounced in NEs with lower HLB_t_, possibly due to a decrease in the surface density of hydrophilic PEG groups.

Furthermore, at a constant SOR and oil ratio, there was a decrease in viscosity and the shear stress/shear rate relationship approaches linearity with increasing HLB_t_ in NEs. This trend can be attributed to the surface density of PEG groups in NEs, which has a significant impact on viscosity and flow behavior. When mixed surfactants with higher HLB_t_ are added to NEs, a larger number of PEG groups are present on the surface of the NEs. This increased presence of PEG groups leads to stronger steric repulsion between nanoparticles, resulting in improved dispersion fluidity, decreased viscosity of NEs, and the manifestation of nearly Newtonian fluid behavior. Similar effects of PEG groups on the flow behavior of nanoscale colloids have been supported by other studies [37,40,54]. 

### 3.6. Encapsulation Efficiency of Cc Oil-NEs for Cur

The encapsulation efficiency of Cc oil-NEs varies with different HLB_t_ values at a constant oil ratio and surfactant amount for Cur, as illustrated in Figure 7. The encapsulation efficiency of Cc oil-NEs for Cur ranged from 72% to 86% and showed a slight increase with higher HLB_t_ values. Since Span 60 and Tween 60 share the same hydrophobic chain consisting of eighteen carbon atoms, the number of PEG headgroups may be a major factor influencing the Cur encapsulation ability of NEs. The presence of a higher quantity of PEG headgroups in NEs with higher HLB_t_ values may provide more opportunities for Cur molecules to be encapsulated. This is supported by the results of DSC and fluorescence polarization measurements, which indicated that NEs with higher HLB_t_ values exhibited greater molecular mobility and a looser molecular arrangement. These characteristics may create more empty spaces within the NEs, facilitating the encapsulation of Cur. Additionally, it was observed that the average size of Cc oil-NEs increased after Cur loading, indicating successful entrapment of Cur within the Cc oil-NEs. Although the specific data for the size increase is not shown, this finding further supports the successful encapsulation of Cur in the NEs.

### 3.7. Chemical Stability of Cur Loaded into Cc Oil-NEs

The protective ability of Cc oil-NEs with different HLB_t_ values for Cur was evaluated by comparing the degradation profiles of Cur activity in NEs to that of a pure Cur dispersion at room temperature. Figure 8 illustrates the percentage of remaining Cur activity over a 12 h period for the pure Cur dispersion and Cur loaded into Cc oil-NEs with various HLB_t_ values. Strikingly, the remaining activity of pure Cur dispersed in a 2% methanol solution decreased to below 20% after 8 h in a sealed sample bottle at room temperature. This significant degradation highlights the low bioavailability of Cur for the human body, as previously reported [3].

In contrast, when Cur was incorporated into Cc oil-NEs, its activity was significantly enhanced to approximately 98% without any decay over a period of 12 h, particularly in NEs with HLB_t_ values of 7.76 or higher. This indicates that the inclusion of Cur into a physically stable formulation of Cc oil-NEs can greatly improve its storage stability. Furthermore, referring to the data presented in Table 1, it is evident that NEs with a HLB_t_ value of 4.6 at SOR = 1 and 4% of oil expressed very short stable time, resulting in a decrease in Cur’s remaining activity to around 80%. This suggests that the degradation rate of Cur in Cc oil-NEs is dependent on their physical storage stability. Overall, the results demonstrate that Cur loaded into Cc oil-NEs with higher HLB_t_ values exhibits improved storage stability and enhanced protection against degradation compared to pure Cur. The physical stability of the NEs plays a crucial role in preserving the activity of Cur during storage.

### 3.8. Cytotoxicity of Cc Oil-NEs 

The in vitro biocompatibility of Cc oil-NEs prepared from Tween 60, Span 60, and Cc oil was assessed using an MTT assay. Figure 9 illustrates the toxic effect of Cc oil-NEs emulsified with a mixture of Span 60 and Tween 60 with different HLB_t_ values on HaCaT cells. The cells were treated with NEs at concentrations ranging from 0.00001 to 0.1 mg/mL. The results show that the cell survival rates remained above 90% for all tested concentrations of Cc oil-NEs, indicating negligible cytotoxic effects on HaCaT cells. This demonstrates the high biocompatibility of these Cc oil-NEs. Similar findings have been reported in previous studies involving other colloidal systems incorporating Span 60 and Tween 60, which also showed low cytotoxicity [55]. Furthermore, the HLB_t_ values of the NEs did not significantly affect the cell viability. However, it should be noted that at the highest concentration of NEs (0.1 mg/mL), a slight decrease in cell survivability was observed. Nevertheless, the overall low cytotoxicity observed suggests the promising potential of these NE formulations for drug delivery applications. The low cytotoxicity observed indicates the great potential of these NE formulations for drug delivery applications. These findings indicate that Cc oil-NEs formulated with Tween 60, Span 60, and Cc oil are biocompatible and have a favorable safety profile, supporting their potential use as drug delivery systems without significant cytotoxic effects on cells.

## 4. Conclusions

In this study, different formulations of Cc oil-NEs were successfully prepared using a combination of homogenization and ultrasonication methods. The characteristics of these NEs, such as size, zeta potential, storage stability, thermotropic phase behavior, intra-particle molecular mobility, and flow behavior, were investigated and found to be influenced by factors such as SOR, HLB_t_ values, and oil ratio. Increasing the addition ratio of Tween 60 in the NEs led to higher HLB_t_ values, resulting in smaller particle size, improved stability, lower phase transition temperature, smaller size of phase transition, higher intra-NE molecular mobility, looser molecular packing, and lower viscosity. Cur was effectively encapsulated into Cc oil-NEs, and the encapsulation efficiency was higher in NEs with higher HLB_t_ values. The incorporation of Cur into physically stable NEs significantly enhanced its storage stability. Moreover, the in vitro biocompatibility of the NE formulations was assessed and found to be excellent. Overall, this study provides valuable insights into the development of Cur carriers by successfully preparing Cc oil-NEs with adjustable physicochemical properties. It also highlights the improved storage stability of Cur when encapsulated in these NEs. These findings contribute to the understanding of NE formulation design and its potential applications in drug delivery systems.

## Figures and Tables

**Figure 1 polymers-15-02864-f001:**
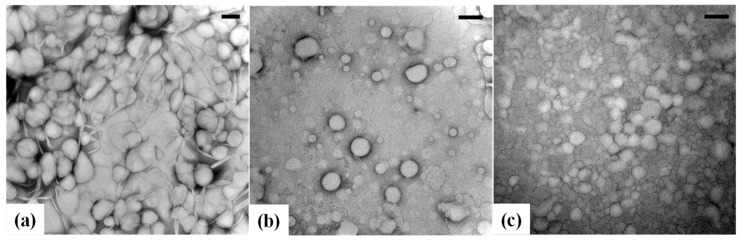
TEM images of Cc oil-NEs emulsified by (**a**) Span 60 (HLB_t_ = 4.7), (**b**) mixed Span 60/Tween 60 (HLB_t_ = 9.8), and (**c**) Tween 60 (HLB_t_ = 14.9) at 4% oil and SOR = 1. The scale bar indicates 200 nm.

**Figure 2 polymers-15-02864-f002:**
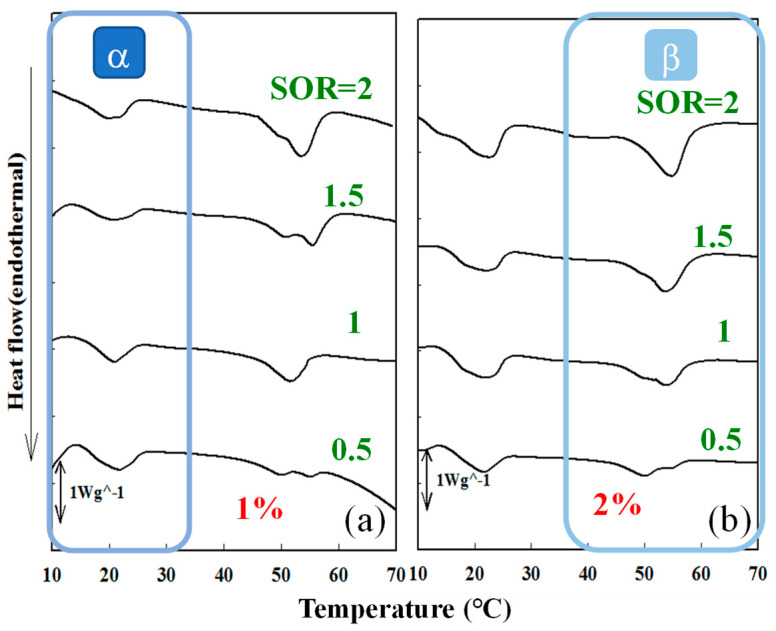
Endothermal DSC curves of Cc oil-NEs containing (**a**) 1% and (**b**) 2% oil emulsified by Span 60 at various SOR values.

**Figure 3 polymers-15-02864-f003:**
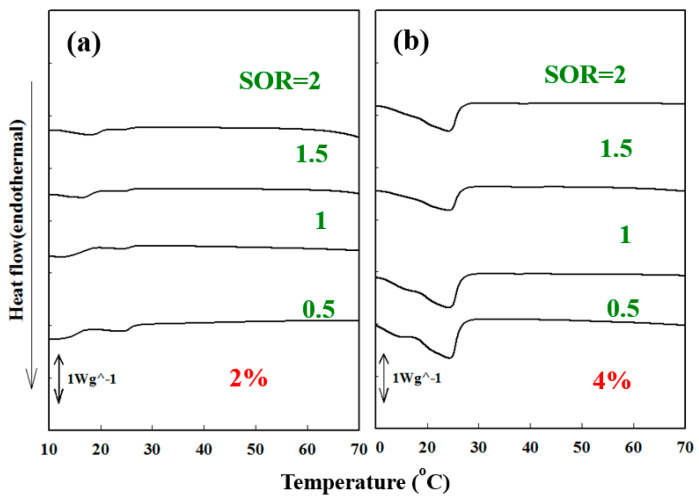
Endothermal DSC curves of Cc oil-NEs with (**a**) 2% and (**b**) 4% oil emulsified by Tween 60 at various SOR values.

**Figure 4 polymers-15-02864-f004:**
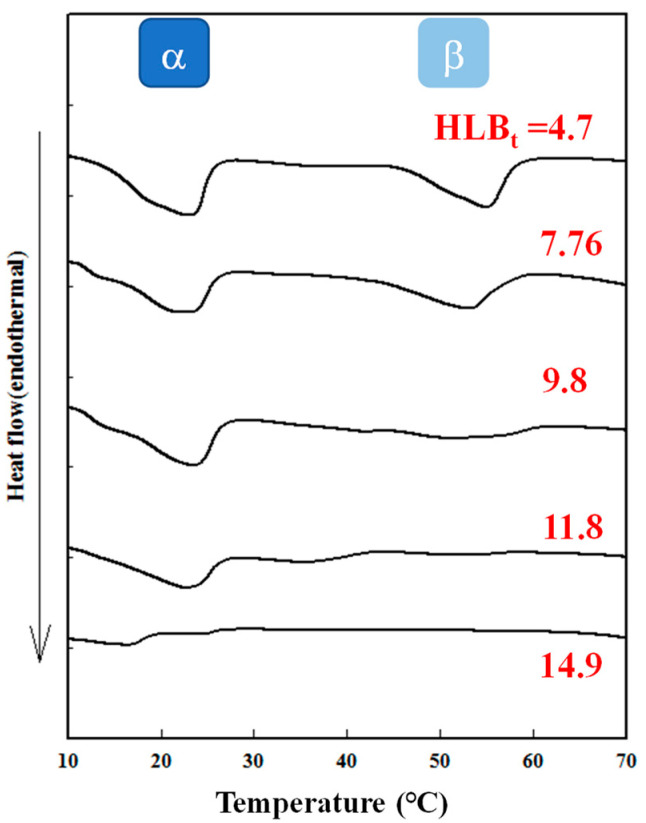
Endothermal DSC curves of Cc oil-NEs emulsified by a combination of Span 60/Tween 60 with varying calculated HLB_t_ values at SOR = 1 and an oil of 2%.

**Figure 5 polymers-15-02864-f005:**
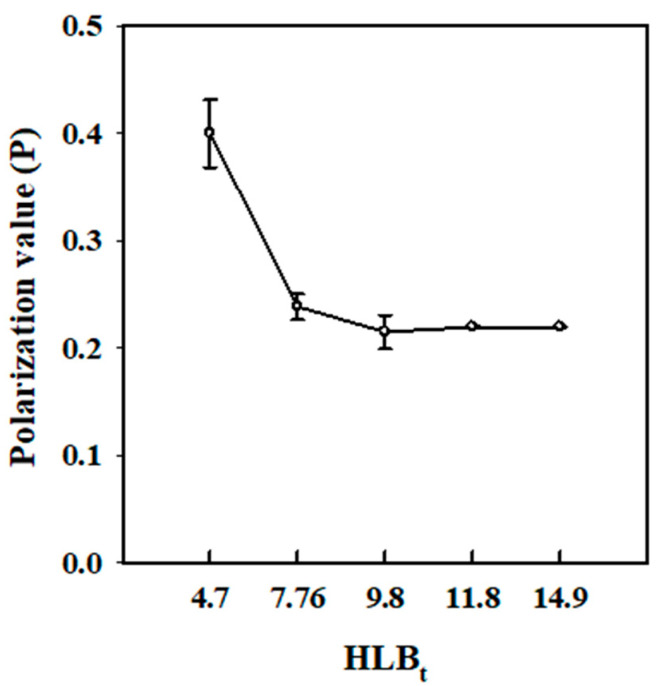
Fluorescence polarization of Cc oil-NEs emulsified by a combination of Span/Tween 60 with varying calculated HLB_t_ values at SOR = 1 and 2% oil.

**Figure 6 polymers-15-02864-f006:**
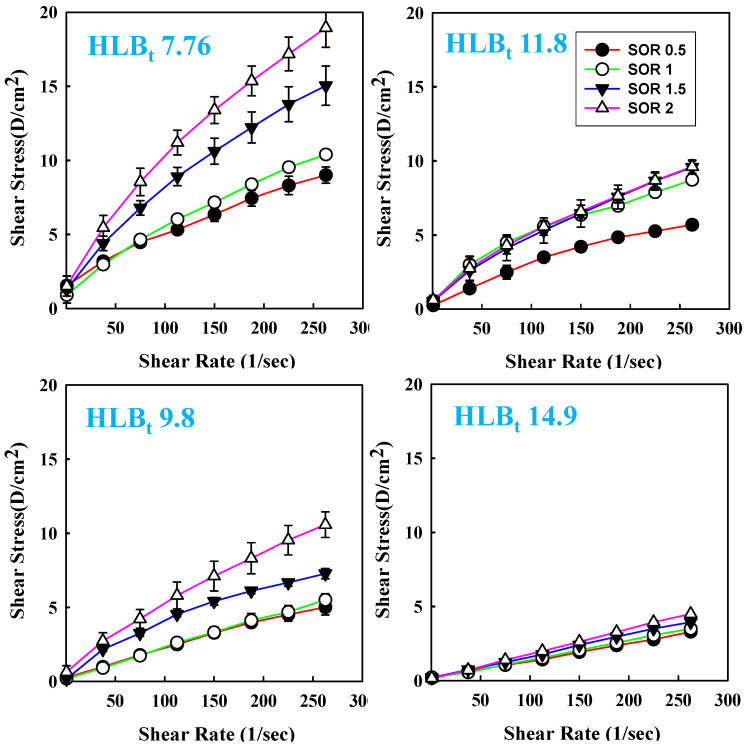
The relationship between shear stress and shear rate of Cc oil-NEs emulsified by a combination of Span/Tween 60 with different calculated HLB_t_ values and various SORs at 2% oil.

**Figure 7 polymers-15-02864-f007:**
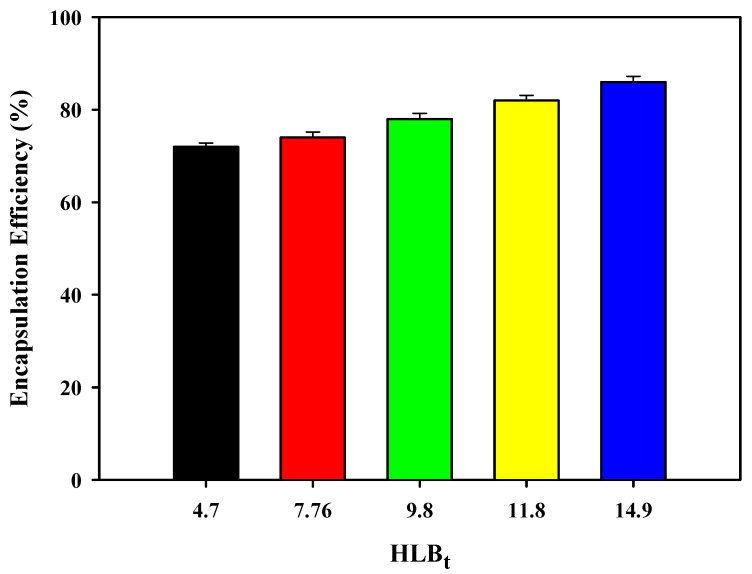
Encapsulation efficiency of Cc oil-NEs emulsified by a mixture of Span 60 and Tween 60 with different HLB_t_ values at SOR = 1 and 2% oil for Cur.

**Figure 8 polymers-15-02864-f008:**
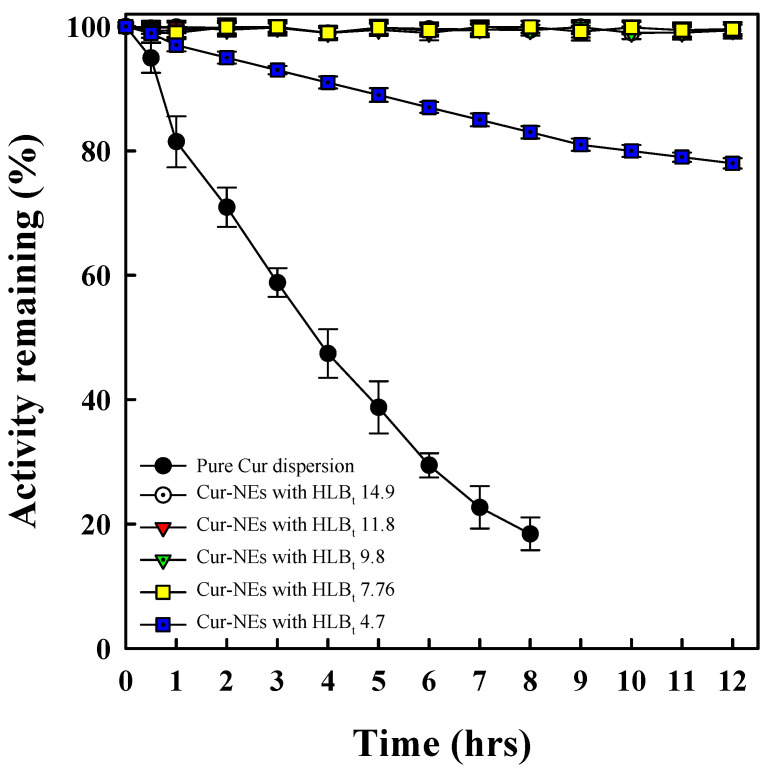
Cur activity comparison between pure Cur dispersion and Cur loaded into Cc oil-NEs with different HLB_t_ values at SOR = 1 and 4% oil over a period of 12 h.

**Figure 9 polymers-15-02864-f009:**
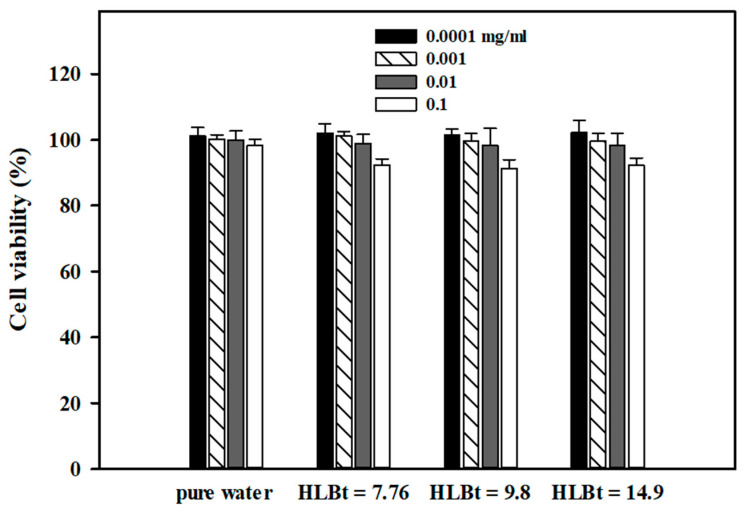
Cytotoxicity of Cc oil-NEs emulsified by a mixture of Span 60 and Tween 60 with different HLB_t_ values toward HaCaT cells. The cells were incubated with different NE concentrations (0.0001, 0.001, 0.01, 0.1 mg/mL) for 24 h, and all NEs were prepared at a fixed SOR of 1 and an oil concentration of 2%. The error bars represent standard deviations calculated from three independent measurements.

**Table 1 polymers-15-02864-t001:** Average particle size (APS) nm, polydispersity (PdI), zeta potential (ZP) mV, and stable days of Cc oil-NEs emulsified by Span 60 at different oil ratios and SORs.

Oil	SOR	APS (nm) ± S.D	PdI ± S.D	ZP (mV) ± S.D	Stable Days
1%	0.5	227.1 ± 9.15	0.206 ± 0.021	−18.6 ± 1.79	12
1%	1	220.4 ± 3.46	0.186 ± 0.001	−22.5 ± 1.75	12
1%	1.5	207.6 ± 6.12	0.197 ± 0.015	−36.8 ± 1.78	8
1%	2	204.4 ± 8.85	0.166 ± 0.025	−28.0 ± 2.47	6
2%	0.5	222.0 ± 2.81	0.214 ± 0.034	−21.8 ± 1.45	7
2%	1	219.6 ± 5.63	0.184 ± 0.019	−23.4 ± 1.01	6
2%	1.5	161.9 ± 48.26	0.188 ± 0.007	−32.6 ± 2.04	6
2%	2	120.9 ± 29.32	0.112 ± 0.099	−29.9 ± 1.89	3
4%	0.5	270.5 ± 14.83	0.194 ± 0.007	−31.9 ± 3.21	1
4%	1	207.9 ± 10.92	0.137 ± 0.028	−33.2 ± 4.74	1
4%	1.5	-	-	-	<1
4%	2	-	-	-	<1
8%	0.5	270.8 ± 13.55	0.255 ± 0.062	−20.4 ± 2.29	<1
8%	1	-	-	-	<1
8%	1.5	-	-	-	<1
8%	2	-	-	-	<1

Each value is shown as mean ± S.D. (*n* = 3).

**Table 2 polymers-15-02864-t002:** Average particle size (APS) nm, polydispersity (PdI), zeta potential (ZP) mV, and stable days of Cc oil-NEs emulsified by Tween 60 at different oil ratios and SORs.

Oil	SOR	APS (nm) ± S.D.	PdI ± S.D.	ZP (mV) ± S.D.	Stable Days
1%	0.5	123.5 ± 5.34	0.237 ± 0.013	−12.7 ± 3.56	346
1%	1	86.7 ± 5.89	0.320 ± 0.014	−7.1 ± 0.88	350
1%	1.5	132.5 ± 11.46	0.256 ± 0.005	−16.7 ± 0.98	344
1%	2	56.2 ± 12.04	0.340 ± 0.017	−19.6 ± 1.03	200
2%	0.5	168.2 ± 9.41	0.239 ± 0.015	−16.5 ± 0.86	350
2%	1	84.9 ± 7.83	0.291 ± 0.016	−13.9 ± 1.87	322
2%	1.5	85.1 ± 3.44	0.337 ± 0.005	−6.0 ± 1.35	360
2%	2	65.4 ± 8.10	0.347 ± 0.008	−5.1 ± 0.94	200
4%	0.5	203.9 ± 10.51	0.268 ± 0.027	−11.2 ± 3.04	300
4%	1	130.8 ± 0.92	0.339 ± 0.006	−9.5 ± 0.25	300
4%	1.5	124.6 ± 11.68	0.300 ± 0.005	−6.7 ± 3.01	252
4%	2	215.1 ± 6.36	0.275 ± 0.016	−13.3 ± 2.65	180
8%	0.5	320.3 ± 71.74	0.089 ± 0.073	−20.8 ± 1.36	220
8%	1	355.4 ± 126.01	0.102 ± 0.099	−16.0 ± 2.30	190
8%	1.5	375.2 ± 51.71	0.323 ± 0.031	−31.6 ± 2.73	180
8%	2	321.7 ± 4.97	0.273 ± 0.019	−19.5 ± 0.61	140

Each value is shown as mean ± S.D. (*n* = 3).

**Table 3 polymers-15-02864-t003:** Average particle size (APS) nm, polydispersity (PdI), zeta potential (ZP) mV, and stable days of Cc oil-NEs emulsified by mixed Span 60/Tween 60 with different HLB_t_ and SORs at 4% oil.

Span 60/Tween 60 (Weight Ratio)	HLB_t_	SOR	APS (nm) ± S.D.	PdI ± SD	ZP (mV) ± S.D.	Stable Days
0/100	14.9	0.5	203.9 ± 10.51	0.268 ± 0.027	−11.2 ± 3.04	300
	14.9	1	130.8 ± 0.92	0.259 ± 0.006	−9.5 ± 0.25	300
	14.9	1.5	124.6 ± 11.68	0.250 ± 0.005	−6.7 ± 3.01	252
	14.9	2	251.7 ± 4.97	0.275 ± 0.016	−13.3 ± 2.65	180
30/70	11.8	0.5	194.0 ± 16.18	0.248 ± 0.036	−32.1 ± 3.23	300
	11.8	1	114.7 ± 3.44	0.222 ± 0.039	−29.2 ± 5.29	295
	11.8	1.5	80.2 ± 15.72	0.278 ± 0.004	−26.8 ± 3.90	270
	11.8	2	208.9 ± 8.04	0.352 ± 0.007	−21.2 ± 5.53	260
50/50	9.8	0.5	197.2 ± 3.61	0.255 ± 0.011	−29.6 ± 1.99	190
	9.8	1	182.8 ± 16.13	0.252 ± 0.007	−29.7 ± 3.27	183
	9.8	1.5	145.9 ± 34.39	0.241 ± 0.029	−24.2 ± 2.27	176
	9.8	2	198.8 ± 6.96	0.249 ± 0.007	−24.8 ± 0.71	111
70/30	7.76	0.5	246.6 ± 14.83	0.241 ± 0.014	−26.1 ± 2.78	35
	7.76	1	244.6 ± 10.33	0.238 ± 0.018	−24.5 ± 3.93	30
	7.76	1.5	271.9 ± 6.09	0.224 ± 0.008	−27.0 ± 7.23	13
	7.76	2	284.6 ± 25.70	0.277 ± 0.038	−34.1 ± 3.76	9
100/0	4.7	0.5	270.5 ± 4.83	0.194 ± 0.007	−31.9 ± 3.21	1
	4.7	1	207.9 ± 10.92	0.137 ± 0.028	−33.2 ± 4.74	1
	4.7	1.5	-	-	-	<1
	4.7	2	-	-	-	<1

Each value is shown as mean ± S.D. (*n* = 3).

## Data Availability

The data presented are available on request from the corresponding author.

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
