# Peer review of "Formulation and Characteristics of Edible Oil Nanoemulsions Modified with Polymeric Surfactant for Encapsulating Curcumin"

_polymers, 2023, doi:10.3390/polym15132864_

Round 1
Reviewer 1 Report
Review of polymers-2435990
Formulation and Characteristics of Edible Oil Nanoemulsions Modified with Polymeric Surfactant for Encapsulating Curcumin
Overall assessment
The manuscript is focused on the encapsulation of curcumin by coconut oil nanoemulsions using the non-ionic surfactants Tween 60 and Span 60 with different oil/surfactant ratios. The physical stability and characterization of the capsules and the nanoemulsions have been studied including, size, polydispersity index, zeta potential, fluorescence polarization and morphology observations. In addition, encapsulation efficiency and chemical stability of curcumin have been analyzed. The manuscript is clear and well written, however there are some deficiencies (detailed below) that should be addressed. I believe these deficiencies can be overcome with a major revisions.
Specific remarks/comments
1. Abstract. Include information about the preference of using Tween 60 against Span 60 as results of this study.
2. Lines 58-62. Include the surfactants used in the referenced study.
3. Lines 75-79. Please include a proper justification for using Span 60 and Tween 60 instead of other non-ionic surfactants. Are these surfactants completely biobased?
4. Line 115-116. Include references to the previous works.
5. Materials and method. Include information about the storage of the samples to analyze the stability, for example Storage Temperature and volume of the sample.
6. Table 3. Include the proportion of both surfactants together with the HLBt.
7. Quality of figures 3 and 4 need to be improved.
8. Some figures only show results for SOR=1 (figure 1, 5 and 7) but figure 4 show results for SOR=1.5, and figure 6, caption indicate SOR=2, but results are for 0.5, 1, 1.5 and 2. Moreover, figures 2 and 3 indicate “at various SOR” but they are not well indicated in the graph. Could the authors clarify about this? or include all the results for all the values of SOR studied in each figure?
9. Please revise typos throughout the manuscript. For example: Lines: 56, 77, 104, 140, 221, 273, 292….
10. Lines 258 and 318. Did you use Tween 80? I think authors want to say Tween 60
Author Response
Response to Reviewer 1
The manuscript is focused on the encapsulation of curcumin by coconut oil nanoemulsions using the non-ionic surfactants Tween 60 and Span 60 with different oil/surfactant ratios. The physical stability and characterization of the capsules and the nanoemulsions have been studied including, size, polydispersity index, zeta potential, fluorescence polarization and morphology observations. In addition, encapsulation efficiency and chemical stability of curcumin have been analyzed. The manuscript is clear and well written, however there are some deficiencies (detailed below) that should be addressed. I believe these deficiencies can be overcome with a major revisions.
Specific remarks/comments
Comment 1
Abstract. Include information about the preference of using Tween 60 against Span 60 as results of this study.
Response 1
Tween 60 and Span 60 are food-grade surfactants approved by the FDA and widely employed in the cosmetic, food, and pharmaceutical industries. We have added this information in the abstract (lines 19-20) and introduction (lines 83-84). Thank you for this suggestion.
Comment 2
Lines 58-62. Include the surfactants used in the referenced study.
Response 2
The reference in lines 58-62 is a review paper that does not focus on specific surfactants. Therefore, we cannot include the surfactants used in the reference in this work. Thank you for this comment.
Comment 3
Lines 75-79. Please include a proper justification for using Span 60 and Tween 60 instead of other non-ionic surfactants. Are these surfactants completely biobased?
Response 3
The related descriptions have been addressed in lines 81-88 and lines 97-102. Although Span 60 and Tween 60 are not completely biobased surfactants, they exhibit excellent safety and can be used in various formulations, including emulsions, creams, lotions, and foams, without causing phase separation or instability. Thank you for this comment.
Comment 4
Line 115-116. Include references to the previous works.
Response 4
The references have been included in lines 128-129. Thank you for this comment.
Comment 5
Materials and method. Include information about the storage of the samples to analyze the stability, for example Storage Temperature and volume of the sample.
Response 5
Many thanks for this suggestion. The information has been addressed in lines 145-146.
Comment 6
Table 3. Include the proportion of both surfactants together with the HLBt.
Response 6
The proportion of Span 60 and Tween 60 has been shown in Table 3.
Comment 7
Quality of figures 3 and 4 need to be improved.
Response 7
Figures 3 and 4 have been replotted as suggested. Thank you for this comment.
Comment 8
Some figures only show results for SOR=1 (figure 1, 5 and 7) but figure 4 show results for SOR=1.5, and figure 6, caption indicate SOR=2, but results are for 0.5, 1, 1.5 and 2. Moreover, figures 2 and 3 indicate “at various SOR” but they are not well indicated in the graph. Could the authors clarify about this? or include all the results for all the values of SOR studied in each figure?
Response 8
Thank you for this comment. In Figure 4, SOR has been revised to 1 due to our typing error. The caption for Figure 6 has been corrected. SOR has been labeled in Figures 2 and 3. However, due to the limitation of sample stability during DSC and fluorescence polarization experiments, not all values of SOR could be shown in the results to ensure data accuracy.
Comment 9
Please revise typos throughout the manuscript. For example: Lines: 56, 77, 104, 140, 221, 273, 292….
Response 9
These typos have been revised as the reviewer’s suggestion. Many thanks this comment.
Comment 10
- Lines 258 and 318. Did you use Tween 80? I think authors want to say Tween 60
Response 10
These typing errors have been revised, and we many thanks for this comment
Please let me know if you need any further assistance.
Sincerely,
Tzung-Han Chou
Professor
Department of Chemical and Materials Engineering
National Yunlin University of Science and Technology
123 University Road, Section 3, Douliou, Yunlin 64002, Taiwan.
Tel: +886-5-5342601-4625
Fax: +886-5-5312071
E-mail: chouth@yuntech.edu.tw
Reviewer 2 Report
Dear Editor,
Thank you for considering our article titled "Recent Advances in Polymer-Based Drug Delivery Systems" for review in Polymers.
After carefully reviewing the article, I would like to provide feedback for major revisions before considering its publication. While the study addresses an important topic and presents intriguing findings, there are several areas that require improvement to enhance the clarity and scientific rigor of the article.
The study provides a brief overview of the study, but it lacks specific details regarding the methods, results, and conclusions. I recommend including key quantitative information, such as MTT and toxicity results.
The introduction should provide a more comprehensive background on the challenges associated with the limited oil solubility and poor stability of curcumin, as well as its potential biomedical applications. Strengthening the introduction with relevant references to support the statements would provide a solid foundation for the study.
The presentation of results and their interpretation should be expanded to provide a comprehensive understanding of the findings. Additionally, a clear discussion should be included to explain the underlying mechanisms responsible for the observed effects, linking the findings to relevant literature and previous studies in the field.
The manuscript would benefit from a thorough proofreading to address grammar, syntax, and clarity issues.
Addressing these major revisions will substantially improve the quality and impact of the article. The revised manuscript should provide a clearer understanding of the experimental methods, a more thorough analysis of the outcomes, and a succinct and insightful conclusion. I am confident that making these changes will significantly increase the manuscript's scientific merit and make it suitable for publication.
Sincerely,
Dear Editor,
Thank you for considering our article titled "Recent Advances in Polymer-Based Drug Delivery Systems" for review in Polymers.
After carefully reviewing the article, I would like to provide feedback for major revisions before considering its publication. While the study addresses an important topic and presents intriguing findings, there are several areas that require improvement to enhance the clarity and scientific rigor of the article.
The study provides a brief overview of the study, but it lacks specific details regarding the methods, results, and conclusions. I recommend including key quantitative information, such as MTT and toxicity results.
The introduction should provide a more comprehensive background on the challenges associated with the limited oil solubility and poor stability of curcumin, as well as its potential biomedical applications. Strengthening the introduction with relevant references to support the statements would provide a solid foundation for the study.
The presentation of results and their interpretation should be expanded to provide a comprehensive understanding of the findings. Additionally, a clear discussion should be included to explain the underlying mechanisms responsible for the observed effects, linking the findings to relevant literature and previous studies in the field.
The manuscript would benefit from a thorough proofreading to address grammar, syntax, and clarity issues.
Addressing these major revisions will substantially improve the quality and impact of the article. The revised manuscript should provide a clearer understanding of the experimental methods, a more thorough analysis of the outcomes, and a succinct and insightful conclusion. I am confident that making these changes will significantly increase the manuscript's scientific merit and make it suitable for publication.
Sincerely,
Author Response
Response to Reviewer 2
After carefully reviewing the article, I would like to provide feedback for major revisions before considering its publication. While the study addresses an important topic and presents intriguing findings, there are several areas that require improvement to enhance the clarity and scientific rigor of the article.
Comment 1
The study provides a brief overview of the study, but it lacks specific details regarding the methods, results, and conclusions. I recommend including key quantitative information, such as MTT and toxicity results.
Response 1
Thank you for your comment. We have addressed this concern by adding the MTT assay and toxicity results in lines 209-222 and lines 444 to 470.
Comment 2
The introduction should provide a more comprehensive background on the challenges associated with the limited oil solubility and poor stability of curcumin, as well as its potential biomedical applications. Strengthening the introduction with relevant references to support the statements would provide a solid foundation for the study.
Response 2
Indeed, the introduction should provide a more comprehensive background. We have already mentioned the challenges associated with limited oil solubility and poor stability of curcumin in lines 49-53. To strengthen the introduction, we have added additional references in lines 53-55, as suggested in the review.
Comment 3
The presentation of results and their interpretation should be expanded to provide a comprehensive understanding of the findings. Additionally, a clear discussion should be included to explain the underlying mechanisms responsible for the observed effects, linking the findings to relevant literature and previous studies in the field.
Response 3
Thank you for your comment. We have revised the results and discussion sections as per your suggestions to provide a more comprehensive understanding of the findings. We have also included a clear discussion explaining the underlying mechanisms responsible for the observed effects and have linked our findings to relevant literature and previous studies in the field.
Comment 4
The manuscript would benefit from a thorough proofreading to address grammar, syntax, and clarity issues.
Response 4
Indeed, we have also identified some typographical errors in the manuscript, and they have been revised as suggested. Furthermore, the entire manuscript has been thoroughly proofread by a native English speaker who is an expert in the field, through a language editing office. We greatly appreciate your comment on this matter.
Please let me know if you need any further assistance.
Sincerely,
Tzung-Han Chou
Professor
Department of Chemical and Materials Engineering
National Yunlin University of Science and Technology
123 University Road, Section 3, Douliou, Yunlin 64002, Taiwan.
Tel: +886-5-5342601-4625
Fax: +886-5-5312071
E-mail: chouth@yuntech.edu.tw
Round 2
Reviewer 1 Report
The authors have adequately addressed the comments, and I think the article is ready for publication.
Reviewer 2 Report
Dear Editor,
I hope this letter finds you well. I am writing to inform you of my decision regarding the manuscript titled " Formulation and Characteristics of Edible Oil Nanoemulsions Modified with Polymeric Surfactant for Encapsulating Curcumin" submitted to Polymers Journal. Having carefully reviewed the manuscript, I am pleased to recommend its acceptance for publication in the journal.
best regards.